# Association between metabolic syndrome and 13 types of cancer in Catalonia: A matched case-control study

Tomàs López-Jiménez[1,2], Talita Duarte-Salles[1], Oleguer Plana-Ripoll[3], Martina Recalde[1,2], Francesc Xavier-Cos[1,4,5,6,7], Diana Puente[1,2]*

1 Fundació Institut Universitari per a la recerca a l'Atenció Primària de Salut Jordi Gol i Gurina (IDIAPJGol), Barcelona, Spain, 2 Universitat Autònoma de Barcelona, Bellaterra, Cerdanyola del Vallès, Spain, 3 Department of Economics and Business Economics, National Centre for Register-based Research, Aarhus University, Aarhus, Denmark, 4 Innovacio•Institut Català de la Salut, Barcelona, Spain, 5 Chairman Primary Care Diabetes Europe, Ekerem, Belgium, 6 Foundation Network of Study Groups of Diabetes in Primary Care (redGDPS), Sabadell, Spain, 7 Primary Care Centre Sant Martí de Provençals, Primary Care Management, Barcelona, Spain

* dpuente@idiapjgol.org

**Data Availability Statement:** In accordance with current European and national law, the data used in this study is only available for the researchers participating in this project. The data and variables

## Abstract

### Background

Metabolic syndrome (MS) is the simultaneous occurrence of a cluster of predefined cardio-vascular risk factors. Although individual MS components are associated with increased risk of cancer, it is still unclear whether the association between MS and cancer differs from the association between individual MS components and cancer. The aim of this matched case-control study was to estimate the association of 13 types of cancer with (1) MS and (2) the diagnosis of 0, 1 or 2 individual MS components.

### Methods

Cases included 183,248 patients ≥40 years from the SIDIAP database with incident cancer diagnosed between January 2008-December 2017. Each case was matched to four controls by inclusion date, sex and age. Adjusted conditional logistic regression models were used to evaluate the association between MS and cancer risk, comparing the effect of global MS versus having one or two individual components of MS.

### Results

MS was associated with an increased risk of the following cancers: colorectal (OR: 1.28, 95%CI: 1.23–1.32), liver (OR: 1.93, 95%CI: 1.74–2.14), pancreas (OR: 1.79, 95%CI: 1.63–1.98), post-menopausal breast (OR: 1.10, 95%CI: 1.06–1.15), pre-menopausal endometrial (OR: 2.14, 95%CI: 1.74–2.65), post-menopausal endometrial (OR: 2.46, 95%CI: 2.20–2.74), bladder (OR: 1.41, 95%CI: 1.34–1.48), kidney (OR: 1.84, 95%CI: 1.69–2.00), non-Hodgkin lymphoma (OR: 1.23, 95%CI: 1.10–1.38), leukaemia (OR: 1.42, 95%CI: 1.31–1.54), lung (OR: 1.11, 95%CI: 1.05–1.16) and thyroid (OR: 1.71, 95%CI: 1.50–1.95). Except for prostate, pre-menopause breast cancer and Hodgkin and non-Hodgkin lymphoma, MS

of this study are obtained from the electronic registries of medical records, which are components of the Information System for Research in Primary Care (SIDIAP) (www.sidiap.org). SIDIAP database and the corresponding research projects were developed thanks to an agreement with the Catalan Health Institute (the owner of the data). Thus, we are not allowed to distribute or make publicly available the data to other parties. However, researchers from public institutions can request data from the SIDIAP and other sources (e.g., Cancer Registries) if they comply with certain requirements. Further information is available online (https://www.sidiap.org/index.php/menu-solicitudes-en/application-proccedure) or by contacting the SIDIAP Team (sidiap@idiapjgol.org).

**Funding:** The project received a research grant from the Carlos III Institute of Health, Ministry of Economy and Competitiveness (Spain), awarded on the 2017 call under the Health Strategy Action 2013–2016 of the National Research Program oriented to Societal Challenges, within the Technical, Scientific and Innovation Research National Plan 2013–2016 (reference PI17/00914), co-funded with European Union ERDF funds (European Regional Development Fund). MR is funded by Wereld Kanker Onderzoek Fonds (WKOF), as part of the World Cancer Research Fund International grant program (grant number:2017/1630).

**Competing interests:** The authors have declared that no competing interests exist.

is associated with a higher risk of cancer than 1 or 2 individual MS components. Estimates were significantly higher in men than in women for colorectal and lung cancer, and in smokers than in non-smokers for lung cancer.

## Conclusion

MS is associated with a higher risk of developing 11 types of common cancer, with a positive correlation between number of MS components and risk of cancer.

## Introduction

Metabolic Syndrome (MS) is the cluster of cardiovascular risk factors such as obesity (specifically central obesity), hypertension, dyslipidaemia and insulin resistance [1]. MS is a growing public health concern due to its high global prevalence. Studies from the United States indicate that MS increases with age and that it has a total prevalence of 24% in the general population and of 50% in patients with ischemic cardiopathy and other cardiovascular conditions [2]. In Spain, the prevalence also increases with age, it ranges between 23% and 31% in the general population, and it affects more men than women in people under 65 years of age [3, 4].

MS was initially considered a risk factor just for cardiovascular disease [5]. However, some studies [6–9] associate MS with a higher risk of liver, colorectal and bladder cancer in men; and endometrial, pancreatic, colorectal, ovarian and post-menopausal breast cancer in women. The results from studies on prostate cancer and MS are inconclusive, while some of them show an increase in risk [10], others show a reduction [5]. A published meta-analysis also found a higher risk of haematological cancer in patients with MS [11].

Some studies show that 1 or 2 components of MS are individually associated with colorectal, breast, endometrial, bladder, kidney, lung and thyroid cancer [9, 12–15]. Specifically, the effect of obesity and diabetes on the incidence of colorectal, pancreatic, liver, kidney, breast and endometrial cancer has already been described [16, 17]. However, no evidence has been yet provided for the impact of MS components in other less common cancers [11]. Large population studies are needed to elucidate if the risk of MS on cancer is higher than the risk associated with each MS component.

The main aim of this study was to investigate the association between MS and 13 types of cancer in Catalonia, using data collected from 2006–2017 in a large electronic health records validated database [18, 19]. We also aimed to evaluate the association of one or two MS components with cancer risk.

## Material and methods

### Data source and setting

We conducted a matched case-control study using the Information System for Research in Primary Care (SIDIAP; www.sidiap.org) [18]. This database comprises the electronic health records of 286 primary healthcare centres (6 million of patients, 80% of residents of Catalonia, Spain). The SIDIAP includes sociodemographic data, clinical diagnoses (using the International Classification of Diseases (ICD-10)), clinical variables, referrals, laboratory tests results and medication invoices (using the Anatomical Therapeutic Chemical (ATC) Classification System).

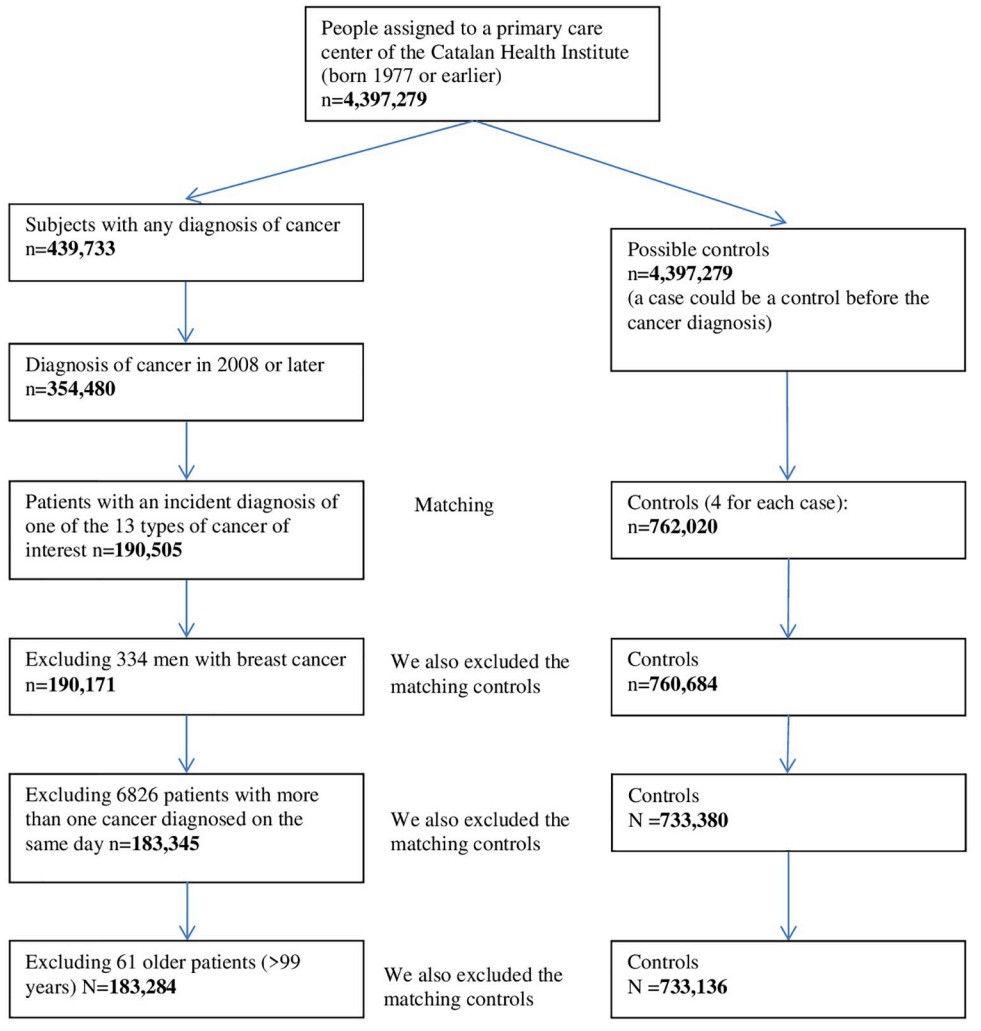

**Fig 1. Study flow chart.**

## Study population

All individuals ≥ 40 years of age with information in the SIDIAP database between 01/01/2006 and 31/12/2017 were suitable to be included. Patients were excluded from participation when they presented with secondary cancers and metastases.

A total of 190,505 individuals with incident cancer were included. Of these, we later excluded 334 men with breast cancer, 6,826 cases because they were diagnosed with more than one cancer on the same day, and 61 because they were >99 years of age on the index date. Finally, a total of 183,284 cases and 733,136 paired controls, four controls for each case, were included (Fig 1).

## Cancer definition

Cancer cases were defined as individuals with an incident diagnosis of selected types of cancer between 01/01/2008 and 31/12/2017. We decided to include the most frequent cancer types (ICD-10 codes) in Spain as outcomes. Even though there is evidence of the association

between MS and some of these cancer types such as colorectal (C18+20), prostate (C61), liver (C22), bladder (C67), endometrium (C54), pancreas (C25) and breast (C50). Prior studies have not investigated the MS-cancer association for several cancer types using a systematic analysis approach like lung cancer (C34) and kidney cancer (C64) but were included due to their high prevalence in the general population. In addition, we included some less frequently occurring cancer types such as thyroid (C73), Hodgkin lymphoma (C81), non- Hodgkin lymphoma (C82-85) and leukemia (C91-95), for which the current literature is limited.

An association between MS and more cancer types than currently recognized in the literature is possible given that the components of MS can trigger biological (hormonal, inflammation, and oxidative stress) processes involved in tumor development.

Breast and endometrial cancers were categorized into pre- and post-menopausal because of the well-established evidence indicating a different impact of obesity and estrogens on these two stages of life [20]. The date of the cancer diagnosis was considered as the index date for cases.

Cancer diagnoses in the SIDIAP are validated against population-based cancer registries [19].

## Control definition

Four controls obtained from the source population were selected for each case, considering as index date of the control the same date of the selection of the case. Each paired case-control was of the same sex and age (± 1 year). No more controls were obtained as it has been previously shown that little statistical power is gained by further increasing this ratio [21].

## MS definition

According to the American Heart Association/National Heart, Lung, and Blood Institute (AHA/NHLBI) criteria, a patient is diagnosed with MS when they present with 3 or more of the following variables: Obesity, High Blood Pressure (HBP), reduced HDL cholesterol, elevated Triglycerides and high Glycemia [1].

Obesity is defined as a body mass index (BMI)> 30 kg/m2, an indicator of overall adiposity. Although central adiposity (usually measured with waist circumference (WC)) is preferred to define this component, we used the BMI in agreement with the WHO definition of MS, since WC was unavailable for most patients in the SIDIAP database [22].

Details for MS construction are published elsewhere [23].

When an abnormal value of any MS component was identified in the database it was assessed the association between cancer and one MS component. If a second component was identified, the association between cancer and two MS component was considered, independently of the time elapsed between the first and the second component identified. When a third component was identified, it was considered that the patient had ≥3 components (and diagnosed with MS). In some patients, more than one measure was recorded on the same day; we considered the average of these values.

Following these definitions, a composite variable of 0, 1, 2, ≥3 MS components was constructed. Both cases and controls had to be exposed either to MS or to 1 or 2 MS components for at least 2 years before the index date (cancer diagnosis or control identification) to avoid reverse causality.

## Covariables

We also extracted information (2 years before the index date) on age; sex (women, men); nationality (Spanish, non-Spanish); the MEDEA deprivation index (census tract-based

deprivation index to identify socioeconomic status in urban areas) categorized in quintiles and rural area; smoking status (non-smokers, ex-smokers and current smokers); alcohol intake calculated in standard units (no alcohol, low and high consumption); dispensation of drugs such as hormonal replacement therapy among menopausal women, paracetamol, aspirin and ibuprofen (classified as yes/ no); presence of hepatitis (classified as yes/ no) and menopause (classified as yes/ no). Women without information on menopausal status ≥ 50 years of age at least two years before the index date were considered to be menopausal.

## Statistical analysis

An initial descriptive analysis of the included population was performed using mean (standard deviation) and median (interquartile range) for quantitative variables and percentages for categorical variables. To assess differences between cases and controls, the t-test or the U Mann-Whitney test for quantitative variables and the Chi-squared test for qualitative variables were performed.

We conducted a conditional logistic regression model to evaluate the association between MS and cancer risk, comparing the effect of global MS versus the individual components of MS, and controlling for the following potential confounders: age, MEDEA Deprivation Index, smoking status and nationality. Hepatitis and other liver diseases were included as confounders in the liver cancer analysis.

All analyses were stratified by type of cancer. Additionally, interaction analyses were performed to explore if the association between MS and cancer differed according to sex and smoking status. To address potential biases due to variables with missing information, multiple imputation by chained equations with 20 imputed datasets was applied to covariates [24–26]. Estimates from each imputed dataset were combined following the rules outlined by Rubin [27].

To assess potential exposure misclassification due the use of BMI instead of WC, we also conducted a sensitivity analysis including only people with at least one WC measurement in the database (WC ≥102 cm in men and ≥88 cm in women are considered central obesity indicators).

Further sensitivity analyses considered two measures of each component separated at least by 2 weeks (maximum 1 year) to ensure that the patient had that component of MS.

The level of statistical significance was 0.05. All analyses were carried out with the statistical packages SPSS 24 (SPSS Inc., Chicago, IL, USA) and Stata 15 (StataCorp LLC., College Station, Texas, USA).

## Ethics approval and consent to participate

This study follows all national and international regulations: Declaration of Helsinki and Principles of Good Research Practice.

In accordance with European and Spanish legislation on confidentiality and data protection ([EU] 2016/679), the data contained in SIDIAP are always pseudonymised. Thus, it is not necessary to ask for informed consent from the participants and so was waived by the Clinical Ethics Committee at IDIAPJGol.

For the link with the CMBD database, SIDIAP uses a third party to ensure confidentiality. The study protocol was approved by the Clinical Research Ethics Committee of IDIAPJGol (P17/212) on November 29, 2017. Anonymity and confidentiality of data and medical records were guaranteed at all times in accordance with the Organic Law 15/1999 on the Protection of Personal Data (http://www.boe.es/boe/dias/1999/12/14/pdfs/A43088-43099.pdf).

## Results

The distribution of cancer in the 183,284 cases was as follows: 36,204 colorectal; 5,754 liver; 5,417 pancreas; 37,647 breast (13,572 pre-menopausal breast and 24,075 post-menopausal breast); 5,386 endometrial (1,124 pre-menopausal endometrial and 4,262 post-menopausal endometrial); 20,799 bladder; 6,833 kidney; 30,888 prostate; 682 Hodgkin lymphoma; 3,621 non-Hodgkin lymphoma; 6,957 leukaemia; 20,387 lung and 2,709 thyroid (Table 1). Four controls for each case (733,136 in total) were selected. Fig 1 shows the flow chart of the selection process of the study participants.

**Table 1. Association between selected cancers and number of Metabolic syndrome components.**

|  | Metabolic Syndrome n(%) | | | |
|---|---|---|---|---|
|  | **0 components** | **1 component** | **2 components** | **MS(≥3 components)** |
| N total | 253661 | 236777 | 175968 | 250014 |
| **Digestive** |  |  |  |  |
| Colorectal Cancer | 7957 (22.0) | 9157 (25.3) | 7394 (20.4) | 11696 (32.3) |
| Controls | 35959 (24.8) | 37848 (26.1) | 28849 (19.9) | 42160 (29.1) |
| Liver Cancer | 1067 (18.5) | 1483 (25.8) | 1239 (21.5) | 1965 (34.2) |
| Controls | 5994 (26.0) | 5967 (25.9) | 4477 (19.5) | 6578 (28.6) |
| Pancreas Cancer | 996 (18.4) | 1257 (23.2) | 1168 (21.6) | 1996 (36.8) |
| Controls | 5252 (24.2) | 5488 (25.3) | 4387 (20.2) | 6541 (30.2) |
| **Gynecological** |  |  |  |  |
| Pre-Menopause Breast Cancer | 8251 (60.8) | 3130 (23.1) | 1243 (9.2) | 948 (7.0) |
| Pre-Menopause Controls | 32397 (60.1) | 11798 (21.9) | 5249 (9.7) | 4480 (8.3) |
| Post-Menopause Breast Cancer | 5552 (23.1) | 6655 (27.6) | 4672 (19.4) | 7196 (29.9) |
| Post-Menopause Controls | 23359 (24.2) | 26665 (27.6) | 18624 (19.3) | 28016 (29.0) |
| Pre-Menopause Endometrial Cancer | 515 (45.8) | 272 (24.2) | 157 (14.0) | 180 (16.0) |
| Pre-Menopause Controls | 2530 (56.7) | 1045 (23.4) | 466 (10.4) | 422 (9.5) |
| Post-Menopause Endometrial Cancer | 647 (15.2) | 942 (22.1) | 842 (19.8) | 1831 (43.0) |
| Post-Menopause Controls | 4074 (23.9) | 4512 (26.4) | 3362 (19.7) | 5133 (30.1) |
| **Urological** |  |  |  |  |
| Bladder Cancer | 4152 (20.0) | 5281 (25.4) | 4484 (21.6) | 6882 (33.1) |
| Controls | 20233 (24.3) | 21474 (25.8) | 17261 (20.7) | 24228 (29.1) |
| Kidney Cancer | 1517 (22.2) | 1756 (25.7) | 1298 (19.0) | 2262 (33.1) |
| Controls | 8012 (29.3) | 6848 (25.1) | 5138 (18.8) | 7334 (26.8) |
| Prostate Cancer | 6725 (21.8) | 8836 (28.6) | 6940 (22.5) | 8387 (27.2) |
| Controls | 29141 (23.6) | 32839 (26.6) | 26110 (21.1) | 35462 (28.7) |
| **Hematological** |  |  |  |  |
| Hodgkin Lymphoma | 291 (42.7) | 141 (20.7) | 116 (17.0) | 134 (19.6) |
| Controls | 1220 (44.7) | 642 (23.5) | 379 (13.9) | 487 (17.9) |
| Non-Hodgkin Lymphoma | 1115 (30.8) | 909 (25.1) | 677 (18.7) | 920 (25.4) |
| Controls | 4895 (33.8) | 3624 (25.0) | 2482 (17.1) | 3483 (24.0) |
| Leukaemia | 1532 (22.0) | 1788 (25.7) | 1443 (20.7) | 2194 (31.5) |
| Controls | 7334 (26.4) | 7157 (25.7) | 5426 (19.5) | 7911 (28.4) |
| **Others** |  |  |  |  |
| Lung Cancer | 5005 (24.5) | 5303 (26.0) | 4050 (19.9) | 6029 (29.6) |
| Controls | 22142 (27.2) | 20694 (25.4) | 16031 (19.7) | 22681 (27.8) |
| Thyroid Cancer | 1002 (37.0) | 663 (24.5) | 461 (17.0) | 583 (21.5) |
| Controls | 4795 (44.3) | 2603 (24.0) | 1543 (14.2) | 1895 (17.5) |

Baseline characteristics of cases and controls are summarized in Table 2. The mean age of cases and controls was 67.5 years (SD 12.4). Women accounted for 56.3% of study participants. Table 1 shows the distribution of the composite variable related to the number of MS components by different types of cancer. An association was observed between the number of MS components and all cancers studied, except for pre-menopausal breast cancer, prostate cancer and Hodgkin Lymphoma. MS prevalence was higher in cases than in controls except for pre-menopausal breast cancer (7.0% vs. 8.3%) and prostate cancer (27.2% vs. 28.4%). The cancer with the highest prevalence of MS was post-menopausal endometrial cancer (43.0% in cases compared to 30.1% in matched controls).

Hypertension was the most frequent component of MS among the patients included in the study with exposure to only one component (80.3 and 80.4 cases and controls, respectively). In patients exposed to two components, the most frequent combination was hypertension + high glycemia (47.1 and 46.3 cases and controls, respectively). Lastly, in patients exposed to ≥3 components, the most frequent combination was hypertension + high glycemia + obesity (17.5 and 17.7 cases and controls, respectively) (S1 Table).

Regarding controls, more MS components (gradient from 0 to ≥ 3) were observed in women, older patients, participants living in deprived areas, smokers and patients with a lower registered consumption of paracetamol, ASA and ibuprofen (S2 Table).

The cancer types associated with MS in the adjusted models were post-menopausal endometrial (OR 2.46, 95%CI 2.20–2.74), pre-menopausal endometrial (OR 2.14, 95%CI 1.74–2.65), liver (OR 1.93, 95%CI 1.74–2.14), kidney (OR 1.84, 95%CI 1.69–2.00), pancreas (OR 1.79, 95%CI 1.63–1.98), thyroid (OR 1.71, 95%CI 1.50–1.85), leukaemia (OR 1.42, 95%CI 1.31–1.54), bladder (OR 1.41, 95%CI 1.34–1.48), colorectal (OR 1.28, 95%CI 1.23–1.32), non-Hodgkin lymphoma (OR 1.23, 95%CI 1.10–1.38), lung (OR 1.11, 95%CI 1.05–1.16) and post-menopausal breast (OR 1.10, 95%CI 1.06–1.15). No association was found between MS and Hodgkin lymphoma (OR 1.19, 95%CI 0.78–1.82). The ORs in gynaecological cancers were higher in post-menopausal (OR: 1.10 95%CI: 1.06–1.15 and OR: 2.46 95%CI: 2.20–2.75 for breast and endometrial cancer, respectively) than pre-menopausal women (OR: 0.85, 95%CI: 0.78–0.92 and OR: 2.14 95%CI: 1.74–2.65 for breast and endometrial cancer, respectively) (Fig 2).

The increasing number of MS components positively correlates with cancer risk in adjusted models, except for prostate, lung, pre-menopausal breast cancer and non-Hodgkin lymphoma. With the increasing number of MS components, the protective power on pre-menopausal breast cancer increase. Interestingly, while MS was not associated with increased risk of prostate cancer, there was a correlation between the presence of 1 or 2 components of MS and the risk of this cancer (OR 1.15, 95%CI 1.11–1.19 and OR 1.14, 95%CI 1.10–1.19 for 1 and 2 components, respectively). In contrast, the risk of lung cancer was similar for participants with 1, 2 and ≥ 3 (MS) components (OR 1.09, 95%CI 1.05–1.15 and OR 1.08. 95%CI 1.02–1.13 for 1 and 2 components and OR 1.11. 95%CI 1.05–1.16 for MS). Participants with 1 or 2 components presented a higher risk of pre-menopausal breast cancer than participants with MS (OR 1.03, 95%CI 0.98–1.08 and OR 0.94, 95%CI 0.88–1.01 for 1 and 2 components and OR 0.85, 95%CI 0.78–0.92 for MS). Participants with 2 components presented a similar risk of non-Hodgkin lymphoma than participants with MS (Fig 2).

We stratified all MS-cancer associations by sex (Fig 3). For colorectal and lung cancer, the risk of MS was higher in men (OR: 1.33 95%CI 1.27–1.40 and OR: 1.14 95%CI 1.08–2.20, respectively) than in women (OR: 1.20 95%CI: 1.14–1.27 and OR: 1.01 95%CI: 0.90–1.12, respectively). The p-values for the interaction between sex and MS were 0.004 and 0.002 for colorectal and lung cancer, respectively. Stratification by sex did not show further differences in the association between MS and cancer.

**Table 2. Characteristics of cancer cases and matched controls.**

| | All Cases n(%) | All Controls n(%) |
|---|---|---|
| N total | 183284 | 733136 |
| **Age** mean (SD) | 67.5 (12.4) | 67.5 (12.4) |
| Median (IQR) | 68 (58–77) | 68 (58–77) |
| **Metabolic Syndrome** | | |
| No component | 46324 (25.3) | 207337 (28.3) |
| 1 component | 47573 (26.0) | 189204 (25.8) |
| 2 components | 36184 (19.7) | 139784 (19.1) |
| MS | 53203 (29.0) | 196811 (26.8) |
| **Sex** | | |
| Men | 80051 (43.7) | 320204 (43.7) |
| Women | 103233 (56.3) | 412932 (56.3) |
| **Nationality** | | |
| Spanish | 178241 (97.2) | 691888 (94.4) |
| Non-Spanish | 5043 (2.8) | 41248 (5.6) |
| **MEDEA index** | | |
| Quintile 1 | 29788 (16.3) | 117426 (16.0) |
| Quintile 2 | 26394 (14.4) | 104654 (14.3) |
| Quintile 3 | 25126 (13.7) | 102756 (14.0) |
| Quintile 4 | 23655 (12.9) | 100601 (13.7) |
| Quintile 5 | 20359 (11.1) | 88566 (12.1) |
| Rural | 33951 (18.5) | 138931 (19.0) |
| Missings | 24011 (13.1) | 80202 (10.9) |
| **Smoking status** | | |
| Never smoker | 58529 (31.9) | 248919 (34.0) |
| Ex-smoker | 20881 (11.4) | 75228 (10.3) |
| Smoker | 19780 (10.8) | 62682 (8.5) |
| Missings | 84094 (45.9) | 346307 (47.2) |
| **Alcohol intake** | | |
| No consumption | 58923 (32.1) | 233338 (31.8) |
| Low consumption | 34357 (18.7) | 127687 (17.4) |
| High consumption | 3561 (1.9) | 10967 (1.5) |
| Missings | 86443 (47.2) | 361144 (49.3) |
| **Hormonal therapy (women postmenopausia)** | | |
| No consumption | 54999 (93.4) | 220858 (93.5) |
| Consumption | 3916 (6.6) | 15250 (6.5) |
| **Paracetamol** | | |
| No consumption | 129196 (70.5) | 530011 (72.3) |
| Consumption | 54088 (29.5) | 203125 (27.7) |
| **Acetylsalicylic acid (ASA)** | | |
| No consumption | 151326 (82.6) | 610683 (83.3) |
| Consumption | 31958 (17.4) | 122453 (16.7) |
| **Ibuprofen** | | |
| No consumption | 156708 (85.5) | 633580 (86.4) |
| Consumption | 26576 (14.5) | 99556 (13.6) |
| **Chronic Hepatitis** | | |
| No hepatitis | 181927 (99.3) | 724036 (98.8) |
| Hepatitis B | 258 (0.1) | 2114 (0.3) |

(*Continued*)

**Table 2.** (Continued)

| | All Cases n(%) | All Controls n(%) |
|---|---|---|
| Hepatitis C | 1078 (0.6) | 6853 (0.9) |
| Other/unspecified hepatitis | 21 (0) | 133 (0) |
| **Menarche age** mean (SD) | 12.6 (1.6) | 12.7 (1.6) |
| Median (IQR) | 13 (12–14) | 13 (12–14) |
| Missings n(%) | 63746 (79.6) | 258683 (80.8) |
| **Menopause** | | |
| No | 21136 (26.4) | 84096 (26.3) |
| Yes | 58915 (73.6) | 236108 (73.7) |
| **Primary care visits between 2 and 4 years before data index** mean (SD) | 14.2 (17.4) | 13.5 (17.6) |
| Median (IQR) | 9 (1–21) | 8 (0–20) |

SD, Standard Deviation; IQR, Inter Quartile Range, MS, Metabolic Syndrome.

The association between MS and lung cancer changed when the analysis was stratified according to smoking status (interaction term p value <0.001); the risk in smokers and ex-smokers was higher than in non-smokers (OR: 1.34 95%CI: 1.18–1.52 in smokers, OR: 1.19 95%CI 1.04–1.35 in ex-smokers and OR: 0.93 95%CI: 0.85–1.01 in non-smokers). (Table 3).

We performed two sensitivity analyses in which we altered the main definition of MS.

In the first analysis, two abnormal measures were used to ensure that the patient was exposed to that component. In a second analysis, we used WC instead of BMI to report obesity. The results in sensitive analyses using two measures to define MS components were similar for almost all cancers. However, for liver, kidney and Hodgkin lymphoma the ORs in the models with two measures were slightly higher than the ORs of the main models. The largest difference was found for kidney cancer (OR: 1.84 95%CI 1.69–2.00 vs. OR: 2.23 95%CI 1.95–2.60, in one and two measures, respectively). The ORs were similar when WC was used instead of BMI, except for prostate and lung cancer, although the sample was small due to the high number of missing values. In the main analysis, MS was not associated with prostate cancer. However, when using WC instead of BMI, MS was inversely associated with prostate cancer (OR: 1.02 95%CI 0.98–1.06 vs. OR: 0.70 95%CI 0.55–0.90, respectively). In contrast, in the main analysis MS was a risk factor for lung cancer and when using WC, MS was not associated with lung cancer (OR: 1.11 95%CI 1.05–1.16 vs. OR: 1.01 95%CI 0.70–1.450, respectively (S3 Table)). The Kappa concordance index between BMI and WC was 0.492.

## Discussion

In this large population-based study, MS was associated with an increased risk of 11 out of 13 cancers, namely endometrial, liver, kidney, pancreas, leukaemia, bladder, colorectal, non–Hodgkin lymphoma, lung and post-menopausal breast, although the effects differed substantially by cancer type. An increasing number of MS components positively correlated with a significant increase in cancer risk in adjusted models, except for non-Hodgkin lymphoma and prostate, lung and pre-menopausal breast cancer. The observed effect sizes for the cancers associated with MS in our data were broadly consistent with previous studies [7, 9, 13–15, 28–34]. Contrary to our study, Park et al. reported a weaker association between MS and thyroid cancer [14], while Almquist and colleagues failed to report any association [35] using a z-score (standard score) calculation that included all 5 components of MS. In agreement with other studies, our results do not show an association between MS and prostate cancer or Hodgkin

| CANCER | MS | | OR | 95%CI |
|---|---|---|---|---|
| **Digestive** | | | | |
| Colorectal | 1 component | | 1.09 | 1.05-1.13 |
| | 2 components | | 1.17 | 1.12-1.21 |
| | MS | | 1.28 | 1.23-1.32 |
| Liver* | 1 component | | 1.33 | 1.20-1.47 |
| | 2 components | | 1.59 | 1.43-1.77 |
| | MS | | 1.93 | 1.74-2.14 |
| Pancreas | 1 component | | 1.26 | 1.15-1.39 |
| | 2 components | | 1.52 | 1.37-1.68 |
| | MS | | 1.79 | 1.63-1.98 |
| **Gynaecological** | | | | |
| Breast | 1 component | | 1.03 | 0.98-1.08 |
| Pre-Menopausal | 2 components | | 0.94 | 0.88-1.01 |
| | MS | | 0.85 | 0.78-0.92 |
| Breast | 1 component | | 1.04 | 1.00-1.09 |
| Post-Menopausal | 2 components | | 1.06 | 1.02-1.11 |
| | MS | | 1.10 | 1.06-1.15 |
| Endometrial | 1 component | | 1.26 | 1.06-1.50 |
| Pre-Menopausal | 2 components | | 1.68 | 1.36-2.08 |
| | MS | | 2.14 | 1.74-2.65 |
| Endometrial | 1 component | | 1.34 | 1.20-1.50 |
| Post-Menopausal | 2 components | | 1.67 | 1.49-1.88 |
| | MS | | 2.46 | 2.20-2.74 |
| **Urological** | | | | |
| Bladder | 1 component | | 1.21 | 1.15-1.27 |
| | 2 components | | 1.29 | 1.22-1.36 |
| | MS | | 1.41 | 1.34-1.48 |
| Kidney | 1 component | | 1.42 | 1.31-1.54 |
| | 2 components | | 1.46 | 1.33-1.59 |
| | MS | | 1.84 | 1.69-2.00 |
| Prostate | 1 component | | 1.15 | 1.11-1.19 |
| | 2 components | | 1.14 | 1.10-1.19 |
| | MS | | 1.02 | 0.98-1.06 |
| **Haematological** | | | | |
| Hodgkin | 1 component | | 0.92 | 0.63-1.34 |
| lymphoma | 2 components | | 1.28 | 0.82-1.99 |
| | MS | | 1.19 | 0.78-1.82 |
| Non-Hodgkin | 1 component | | 1.12 | 1.01-1.24 |
| lymphoma | 2 components | | 1.24 | 1.10-1.39 |
| | MS | | 1.23 | 1.10-1.38 |
| Leukaemia | 1 component | | 1.23 | 1.14-1.33 |
| | 2 components | | 1.34 | 1.22-1.46 |
| | MS | | 1.42 | 1.31-1.54 |
| **Others** | | | | |
| Lung | 1 component | | 1.09 | 1.05-1.15 |
| | 2 components | | 1.08 | 1.02-1.13 |
| | MS | | 1.11 | 1.05-1.16 |
| Thyroid | 1 component | | 1.29 | 1.15-1.45 |
| | 2 components | | 1.57 | 1.38-1.80 |
| | MS | | 1.71 | 1.50-1.95 |

**Fig 2. Adjusted ORs and 95% confidence intervals according to metabolic syndrome by selected cancers.** ORs are presented by squares, with their 95% CIs as horizontal lines; OR, odds ratio; CI, confidence interval. Reference category is 0 components. All models are adjusted by age, MEDEA Deprivation Index, smoking status and nationality. *Also adjusted by hepatitis and others liver diseases. Multiple imputation by chained equations with 20 imputed datasets were applied to outcomes and covariates.

lymphoma [36]. The results from studies on prostate cancer and MS are inconclusive; some studies report a reduced risk [5, 12] and others report an increased risk [10, 37] while our study found no significant risk of prostate cancer associated with MS. These discrepancies might be explained by Hammarsten's hypothesis [38] that MS inversely correlates with local-ised prostate cancer and positively with advanced disease. Furthermore, a study of Gomez-

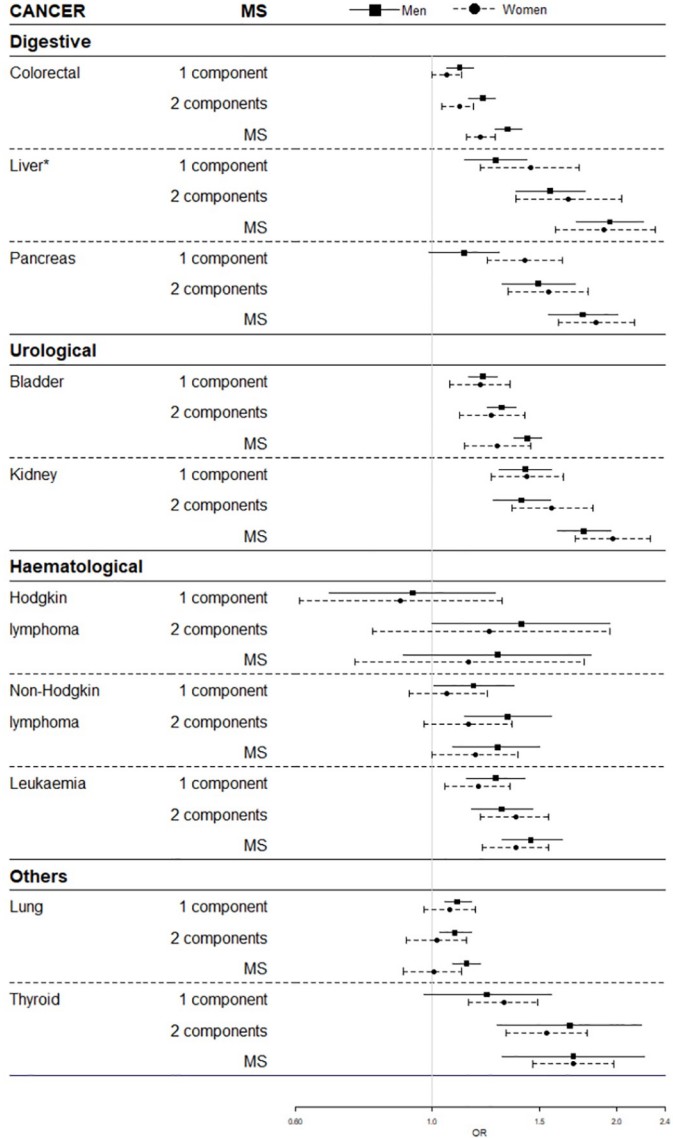

**Fig 3. Adjusted ORs and 95% confidence intervals according to metabolic syndrome by selected cancers and sex.**
ORs are presented by squares (in men) and circles (in women), with their 95% confidence intervals as horizontal lines;
OR, odds ratio. Reference category is 0 components. All models are adjusted by age, MEDEA Deprivation Index,
smoking status and nationality. *Also adjusted by hepatitis and others liver diseases. Multiple imputation by chained
equations with 20 imputed datasets were applied to outcomes and covariates.

Gomez et al., showed that each of the individual criterion of MS, circulating testosterone levels
and inflammatory status may have on the risk and aggressiveness of prostate cancer [10]. In
the case of gynaecological cancers, menopausal status was a determinant factor, especially in
breast cancer. In agreement with previous studies, we observed that MS increased breast can-
cer risk in post-menopausal women, and decreased it in pre-menopausal women [7, 29]. Previ-
ous investigations proposed that each component of the metabolic syndrome is connected
with systemic alterations. Concerning breast cancer, it has been proposed that components of
MS, especially obesity, play different roles in cancer risk according to menopausal status and
estrogen receptor status [39]. Obesity is associated with decreased risk of estrogen receptor–

**Table 3. Adjusted ORs of metabolic syndrome and lung cancer according to tobacco consumption.**

| | Non-smoker | | | Ex-smoker | | | Smoker | | |
|---|---|---|---|---|---|---|---|---|---|
| | OR | 95% CI | P-value | OR | 95% CI | P-value | OR | 95% CI | P-value |
| **Lung cancer in general** | | | | | | | | | |
| No components | 1.00 | | 0.099[1] | 1.00 | | 0.033[1] | 1.00 | | <0.001[1] |
| 1 component | 0.97 | 0.90–1.05 | 0.042[2] | 1.16 | 1.02–1.31 | 0.026[2] | 1.25 | 1.11–1.41 | <0.001[2] |
| 2 components | 0.91 | 0.84–1.00 | | 1.15 | 1.01–1.32 | | 1.31 | 1.15–1.50 | |
| MS | 0.93 | 0.85–1.01 | | 1.19 | 1.04–1.35 | | 1.34 | 1.18–1.52 | |

OR, odds ratio; CI, confidence interval.

Models adjusted by age, medea, alcohol and nationality

[1]Wald test.

[2]P-Trend.

Multiple imputation by chained equations with 20 imputed datasets were applied to outcomes and covariates.

positive breast cancer in premenopausal women, but it is closely related with increased risk of estrogen receptor–positive breast cancer in postmenopausal women [7, 29, 40].

The risk of most cancers was higher in individuals with MS than in patients with one or two components of MS. In agreement with the literature [9, 13–15, 28, 31, 34], a positive correlation between MS components and risk of cancer was found in eight of these eleven cancers (colorectal, liver, pancreas, post-menopausal breast, pre- and post-menopausal endometrial, bladder, leukaemia, and thyroid).

Mechanisms that link metabolic syndrome and cancer risk are not fully understood. Metabolic syndrome may be a surrogate marker for other cancer risk factors, such as decreased physical activity, consumption of high–calorie dense foods, high dietary fat intake, low fiber intake, and oxidative stress [7].

In accordance with previous studies, when stratifying by gender, the risk of colorectal, lung and bladder cancer was higher in men [7, 9, 15, 30]. The positive association between MS and lung cancer was greater in smokers, corroborating the results reported in a recent cohort study [15].

While MS has multiple definitions, the most widely recognised criteria to construct MS belong to the NCEP ATP III and the IDF [1]. A study by Qiao et al. found similar results when comparing NCEP ATP III and IDF criteria for the association between MS and lung cancer [41, 42]. In contrast, Xiang and colleagues found that MS was a risk factor for breast cancer following IDF criteria, whereas no statistical association could be established using NCEP ATP III criteria [39].

We also conducted a sensitivity analysis, in which we required two abnormal measurements of each component for diagnosis, and found that it did not significantly affect the results. When using WC in the analysis instead of BMI, the ORs obtained were similar except for lung and prostate cancer. However, the Kappa index between both measurement methods was low. Since we only had WC measurements for 15% of the population and the number of missing values of this variable was too high, we used BMI criteria for obesity in agreement with other publications [5, 31, 34]. Furthermore, when Montella et al. performed similar sensitivity analyses, their results did not significantly change [9], and Gomez-Gomez et al. found a strong correlation between BMI and WC [10].

Our study has several strengths. Firstly, we used a large data source with sufficient statistical power to investigate associations of less frequent cancers (i.e. Hodgkin and non-Hodgkin lymphoma and thyroid). Consequently, we have been able to present the first results on the

association between MS and haematological malignancies. SIDIAP patients broadly represent the wider population, suggesting good generalizability to the Catalonian and similar populations. In 2019, Recalde et al. [19] validated the diagnosis of cancer in the SIDIAP and the result was that the SIDIAP includes 76% of cancers recorded in the cancer registries [43–45].

Our study also has limitations. Firstly, we assumed that once a person had an abnormal result, this person was constantly exposed to this component even if later results showed improvement. Some evidence points at the concept of metabolic memory, i.e., even when an individual stops meeting MS criteria, they are still at higher risk of specific cancers (i.e. kidney cancer) [13]. Also, the lack of data on other possible confounders influencing the relationship between MS and cancer, such as physical activity and parity, or information related to previous treatment for other health conditions might have biased the results. In addition, it is necessary to explore the potential association of specific MS criteria and risk of specific cancer types in future studies. This is a case-control study, when estimating the association between MS and cancer; however, we were not able to estimate cumulative incidences or other types of absolute risks, which would have been useful to put the relative increase in absolute terms. The tobacco and alcohol variables have a high percentage of missing values (45.9% and 47.2% for tobacco and alcohol, respectively). This is a significant limitation to our study which we have attempted to mitigate through multiple imputation. While it's true that multiple imputation has its own set of biases, current theory suggests that the multiple imputation bias is smaller than the analysis with completed-cases. Considering only the complete-cases of the database would result in a smaller sample size, loss of statistical power and theoretically with more bias [46].

The increasing prevalence of MS worldwide and the high incidence of some cancers suggests that a large number of cancer cases diagnosed every year are related to the metabolic syndrome. There is a compelling need for evidence on whether effective interventions to reduce the prevalence of metabolic syndrome in adult populations could reduce cancer risk. The formulation of public health strategies based on lifestyle changes could obtain significant results in the fight against cancer. Investigating the role of MS as a risk factor of specific cancers is crucial to diagnose and treat cancer in earlier stages.

In summary, MS is associated with a higher risk of developing at least 11 cancer types. The risk of most cancers increased with the number of MS components present in an individual. Our results indicate that prevention strategies targeting individual components of MS could reduce the risk of several cancer types.

## Supporting information

**S1 Table. MS components combination with cancer cases and matched controls.** (DOCX)

**S2 Table. Characteristics of the controls according to metabolic syndrome.** (DOCX)

**S3 Table. Analysis of those patients exposed to a single value of the pathological component versus those who present 2 measures and analysis considering waist circumference instead of BMI.** Adjusted ORs and 95% CI. Multiple imputation by chained equations with 20 imputed datasets were applied to outcomes and covariates. Models adjusted by age, medea, tobacco, alcohol, nationality. [a]Consider two measures of parameters separated at least by 2 weeks (maximum 1 year) to ensure that the patient has that pathological component of MS. [b]Using Waist circumference instead of one measure of BMI. There were 157,872 (86.1%) missing values in the case group and 636,984 (86.9%) missing values in the control group. [c]Also adjusted by hepatitis and others liver disease. [d]The model does not converge when using Waist

circumference instead of one measure od BMI, because the number of observations in this model are very low. [1]Wald test. [2]P-Trend. OR, odds ratio; CI, confidence interval. (DOCX)

## Acknowledgments

We would like to thank the Catalan Institute of Health and the SIDIAP, which provided the database for the study. We would like to thank Eulàlia Farré for the English proofreading of the manuscript.

## Author Contributions

**Conceptualization:** Tomàs López-Jiménez, Talita Duarte-Salles, Diana Puente.

**Data curation:** Tomàs López-Jiménez, Diana Puente.

**Formal analysis:** Tomàs López-Jiménez.

**Funding acquisition:** Diana Puente.

**Investigation:** Tomàs López-Jiménez, Diana Puente.

**Methodology:** Tomàs López-Jiménez, Diana Puente.

**Project administration:** Diana Puente.

**Resources:** Diana Puente.

**Software:** Tomàs López-Jiménez.

**Supervision:** Oleguer Plana-Ripoll, Diana Puente.

**Validation:** Tomàs López-Jiménez, Oleguer Plana-Ripoll, Diana Puente.

**Visualization:** Diana Puente.

**Writing – original draft:** Tomàs López-Jiménez, Diana Puente.

**Writing – review & editing:** Tomàs López-Jiménez, Talita Duarte-Salles, Oleguer Plana-Ripoll, Martina Recalde, Francesc Xavier-Cos, Diana Puente.

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
