## [Decision Letter · Decision Letter 0]

7 Dec 2021

PONE-D-21-30858Association between Metabolic Syndrome and 13 types of Cancer in Catalonia: a matched case-control studyPLOS ONE

Dear Dr. Puente,

Thank you for submitting your manuscript to PLOS ONE. After careful consideration, we feel that it has merit but does not fully meet PLOS ONE’s publication criteria as it currently stands. Therefore, we invite you to submit a revised version of the manuscript that addresses all the points that have been raised by two experts in the field during the review process. Please submit your revised manuscript by January 21, 2022. If you will need more time than this to complete your revisions, please reply to this message or contact the journal office at plosone@plos.org. Please include the following items when submitting your revised manuscript:A rebuttal letter that responds to each point raised by the academic editor and reviewer(s). You should upload this letter as a separate file labeled 'Response to Reviewers'.A marked-up copy of your manuscript that highlights changes made to the original version. You should upload this as a separate file labeled 'Revised Manuscript with Track Changes'.An unmarked version of your revised paper without tracked changes. You should upload this as a separate file labeled 'Manuscript'.

We look forward to receiving your revised manuscript.

Kind regards,

Raul M. Luque, PhD

Academic Editor

PLOS ONE

Journal Requirements:

a) Did participants provide their written or verbal informed consent to participate in this study?

Reviewers' comments:

Reviewer's Responses to Questions

**Comments to the Author**

1. Is the manuscript technically sound, and do the data support the conclusions?

Reviewer #1: Yes

Reviewer #2: Yes

2. Has the statistical analysis been performed appropriately and rigorously? 

Reviewer #1: Yes

Reviewer #2: Yes

3. Have the authors made all data underlying the findings in their manuscript fully available?

Reviewer #1: Yes

Reviewer #2: Yes

4. Is the manuscript presented in an intelligible fashion and written in standard English?

Reviewer #1: Yes

Reviewer #2: Yes

5. Review Comments to the Author

Reviewer #1: This article reported results MS is statistically associated with a higher risk of developing at 11 cancer types after the evaluation of 13. Moreover, they demonstrated in this manuscript that the risk of most cancers increased with the number of MS components present in an individual. Hence, the authors finally suggest that prevention strategies targeting individual components of MS could reduce the risk of several cancer types. Although the manuscript is potentially interesting, it is mostly descriptive and most of the conclusions have been published previously (see comments below).

Thus, there are major aspects that could be revised to improve the results:

- The authors should take care in the use of “recent” when they are referring to other published works since I cannot consider a recent work that has been published more than 5 years ago. An example is given, “MS was initially considered a risk factor just for cardiovascular disease. However, recent studies associate MS with a higher risk of liver, colorectal, and bladder cancer in men; and endometrial, pancreatic, colorectal, ovarian, and postmenopausal breast cancer in women”. However, the original manuscript was published in 2012(doi: 10.2337/dc12-0336).

- The authors did not give too many reasons for the selection of these 13 cancer types. The authors only said, “However, no evidence has been yet provided for the impact of MS components in other less common cancers”. Conversely, for example, lung cancer has related to some metabolic factors of metabolic syndrome. Please, I am really encouraged to provide more reasons for the selection of these cancer types.

- The author must justify the fact of comparing every case with four controls since a major number of control cases could imply a major statistical power being really important for the final results and conclusions of this study.

-In general, the manuscript needs to get a significant improvement in the discussion section. For instance, the sentence is too descriptive: “In gynecological cancers, the menopausal status was a determinant factor, especially in breast cancer. In agreement with previous studies, we observed that MS increased breast cancer risk in postmenopausal women and decreased it in premenopausal women”.

- Concerning the important limitation of not following up with the patients, the authors should clearly explain this limitation and how it could affect the results of the paper. Furthermore, the authors should provide information about when the MS component was identified such as previous to tumor diagnose (1 year or 2 years), during tumor treatment…

Minor comments:

- A sentence in the abstract is not very clear. “Adjusted conditional logistic regression models were used to estimate OR and 95% CI for the association between individual components of MS and cancer, and MS and cancer.” It seems to be confusing.

- The use of “approximately” when you are talking about the patients available in the study is not accurate for scientific work.

- The authors described in methodology “All individuals ≥ 40-99 years of age with information in the SIDIAP database between 01/01/2006 and 31/12/2017 were included. Final participants included patients with any of the 13 types of incident cancers of interest, together with their paired controls.” Please, the authors should provide the number of patients after the application of each inclusion criteria in the methodology section.

- Typing errors like “MS component..” should be corrected in the new manuscript.

Reviewer #2: In this study, Diana Puente et al studied the potential association between Metabolic syndrome (and number of components met) and the risk of 13 types of cancer. The authors included 183,248 patients from the Information System for Research in Primary Care. The data derived from this study showed that Metabolic Syndrome is associated to 11 cancer types (i.e., endometrial, liver, kidney, pancreas, thyroid, leukaemia, bladder, colorectal, non-Hodgkin lymphoma, lung and post-menopausal breast). These results are interesting for the field and shed light on the relation between Metabolic Syndrome and cancer risk. Therefore, the Reviewer consider that this article is well written, adds to the field valuable information and is suitable to be published in PLOS One. However, the following minor comments should be addressed:

- The manuscript is well-written. However, some typos can be found in the text and should be corrected:

o Remove intro in line 130.

o Remove intro in line 131.

o Two period symbols in line 133.

- In the introduction there is no information about the inconclusive relation between prostate cancer and metabolic syndrome. Authors should include some information about that in the introduction.

- In addition to the presence of Metabolic Syndrome, the authors also analysed the impact of the number of criteria met on cancer risk. However, no information was shared with regards to the potential association of specific MS criteria and cancer risk. As an example, Gomez-Gomez et al showed that high-blood pressure was associated with clinically significant prostate cancer in their cohort of patients (1). Authors should analyse this or add some information about this in the limitations paragraph.

- The authors performed two sensitivity analyses in which they altered the main definition of Metabolyc Sindrome. The data obtained when using WC instead of BMI are especially interested as WC is a better criterion to ‘measure’ obesity. Although the authors appreciated that the sample is smaller when classifying the patients using WC due to the high number of missing values, this information (number of patients/missing values) is not showed in Supplemental Table 3. Therefore, authors should include this information in the final version of the manuscript.

- The fact that patients with 1 or 2 MS components presented a higher risk of pre-menopausal breast cancer and prostate cancer than participants with MS has not been discussed in the manuscript. Could the authors explain this rare phenomenon? The authors should address this question in the discussion section of the final manuscript.

References

1. Gomez-Gomez E, Carrasco-Valiente J, Campos-Hernandez JP, Blanca-Pedregosa AM, Jimenez-Vacas JM, Ruiz-Garcia J, Valero-Rosa J, Luque RM, Requena-Tapia MJ. Clinical association of metabolic syndrome, C-reactive protein and testosterone levels with clinically significant prostate cancer. J Cell Mol Med. 2019;23(2):934-942.

6. PLOS authors have the option to publish the peer review history of their article (what does this mean?). If published, this will include your full peer review and any attached files.

Reviewer #1: **Yes: **Fuentes-Fayos, Antonio C

Reviewer #2: No

---

## [Author Response · Author response to Decision Letter 0]

31 Jan 2022

Dear editor,

Many thanks for allowing us the opportunity to revise and resubmit this manuscript. The reviewers have provided us with many constructive suggestions. We have addressed each reviewer’s point below. Two versions of the revised manuscript have been uploaded (a marked-up copy that highlights changes made to the original version, and an unmarked version without tracked changes).

Editorial and formatting comments

1. Please amend your current ethics statement to address the following concerns:

a) Did participants provide their written or verbal informed consent to participate in this study?

RESPONSE: The database is comprised of electronic health records from 286 primary healthcare centres. According to the European and Spanish legislation, it is not necessary to ask for informed consent from the participants to use their data for research after anonymization. Thus, the participants did not provide written or verbal informed consent to participate in this study. The study protocol was approved by the Clinical Research Ethics Committee of the IDIAPJGol (P17/212).

We have changed the “Ethics approval and consent to participate” section (page 8-9, lines, 199-211 in the clean copy). The revised statement is: 

“This study follows all national and international regulations: Declaration of Helsinki and Principles of Good Research Practice.

In accordance with European and Spanish legislation on confidentiality and data protection ([EU] 2016/679), the data contained in SIDIAP are always pseudonymised. Thus, it is not necessary to ask for informed consent from the participants and so was waived by the Clinical Ethics Committee at IDIAPJGol. 

For the link with the CMBD database, SIDIAP uses a third party to ensure confidentiality. The study protocol was approved by the Clinical Research Ethics Committee of IDIAPJGol (P17/212) on November 29, 2017. Anonymity and confidentiality of data and medical records were guaranteed at all times in accordance with the Organic Law 15/1999 on the Protection of Personal Data (http://www.boe.es/boe/dias/1999/12/14/pdfs/A43088-43099.pdf)”.

2. In your Data Availability statement, you have not specified where the minimal data set underlying the results described in your manuscript can be found. PLOS defines a study's minimal data set as the underlying data used to reach the conclusions drawn in the manuscript and any additional data required to replicate the reported study findings in their entirety. All PLOS journals require that the minimal data set be made fully available. For more information about our data policy, please see http://journals.plos.org/plosone/s/data-availability

RESPONSE: We do not have a license to publish the databases by the regulation of the Catalan Health Institute. We have changed the “Data Availability Statement” section (pages 21, lines 438-450 in the clean copy). The revised statement is:

‘In accordance with current European and national law, the data used in this study is only available for the researchers participating in this project. The data and variables of this study are obtained from the electronic registries of medical records, which are components of the Information System for Research in Primary Care (SIDIAP) (www.sidiap.org). SIDIAP database and the corresponding research projects were developed thanks to an agreement with the Catalan Health Institute (the owner of the data). Thus, we are not allowed to distribute or make publicly available the data to other parties. However, researchers from public institutions can request data from the SIDIAP and other sources (e.g., Cancer Registries) if they comply with certain requirements. Further information is available online (https://www.sidiap.org/index.php/menu-solicitudes-en/application-proccedure) or by contacting the SIDIAP Team (sidiap@idiapjgol.org)’

RESPONSE: Please see response to comment 2

RESPONSE: We have included the new ethics statement in the ‘Methods’ section (page 8-9, lines, 199-211 in the clean copy).

RESPONSE: In the new ‘Data Availability Statement’ we explain in detail the ethical or legal restrictions that apply to publicly sharing our data.

Reviewer #1:

1. This article reported results MS is statistically associated with a higher risk of developing at 11 cancer types after the evaluation of 13. Moreover, they demonstrated in this manuscript that the risk of most cancers increased with the number of MS components present in an individual. Hence, the authors finally suggest that prevention strategies targeting individual components of MS could reduce the risk of several cancer types. Although the manuscript is potentially interesting, it is mostly descriptive and most of the conclusions have been published previously (see comments below).

RESPONSE: Thank you for the positive comments. We believe our manuscript provides novel contributions to the scientific community and we show this in our specific responses below.

2. Thus, there are major aspects that could be revised to improve the results:

- The authors should take care in the use of “recent” when they are referring to other published works since I cannot consider a recent work that has been published more than 5 years ago. An example is given, “MS was initially considered a risk factor just for cardiovascular disease. However, recent studies associate MS with a higher risk of liver, colorectal, and bladder cancer in men; and endometrial, pancreatic, colorectal, ovarian, and postmenopausal breast cancer in women”. However, the original manuscript was published in 2012(doi: 10.2337/dc12-0336).

RESPONSE: We agree with the reviewer. We have modified the sentence to exclude the word “recent” (page 3, line 69 and page 3 line73 in the clean copy).

3. The authors did not give too many reasons for the selection of these 13 cancer types. The authors only said, “However, no evidence has been yet provided for the impact of MS components in other less common cancers”. Conversely, for example, lung cancer has related to some metabolic factors of metabolic syndrome. Please, I am really encouraged to provide more reasons for the selection of these cancer types.

RESPONSE: We agree with the reviewer and have modified the paragraph accordingly. The new paragraph is: “We decided to include the most frequent cancer types (ICD-10 codes) in Spain as outcomes. Even though there is evidence of the association between MS and some of these cancer types such as colorectal (C18+20), prostate (C61), liver (C22), bladder (C67), endometrium (C54), pancreas (C25) and breast (C50). Prior studies have not investigated the MS-cancer association for several cancer types using a systematic analysis approach like lung cancer (C34) and kidney cancer (C64) but were included due to their high prevalence in the general population. In addition, we included some less frequently occurring cancer types such as thyroid (C73), Hodgkin lymphoma (C81), non- Hodgkin lymphoma (C82-85) and leukemia (C91-95), for which the current literature is limited. An association between MS and more cancer types than currently recognized in the literature is possible given that the components of MS can trigger biological (hormonal, inflammation, and oxidative stress ) processes involved in tumor development” (page 5, lines 109-121 in the clean copy).

4. The author must justify the fact of comparing every case with four controls since a major number of control cases could imply a major statistical power being really important for the final results and conclusions of this study.

RESPONSE: In case-control studies, the inclusion of more than four or five controls per case provides very little gain in statistical power (Henness S et al.). Thus, we decided to include only four controls per case given that the dataset is already very large and requires a lot of computing power to run the analyses. We included this reference to justify the number of controls. In the ‘Control definition’ section, we add the phrase ‘No more controls were obtained as it has been previously shown that little statistical power is gained by further increasing this ratio (ref)’ (page 6, lines 132-134 in the clean copy)

Reference included: Hennessy S, Bilker WB, Berlin JA et al. Factors influencing the optimal control-to-case ratio in matched case-control studies. Am J Epidemiol. 1999 15;149(2):195-7. 

5. In general, the manuscript needs to get a significant improvement in the discussion section. For instance, the sentence is too descriptive: “In gynecological cancers, the menopausal status was a determinant factor, especially in breast cancer. In agreement with previous studies, we observed that MS increased breast cancer risk in postmenopausal women and decreased it in premenopausal women”.

RESPONSE: This is an epidemiological observational study and we tried to be cautious in making any conclusions that could refer to causality. One of our objectives was to provide evidence on the association between MS and cancer, but we need to be careful when making interpretations of such associations. However, considering the reviewer's comment, we tried to include further details in this specific sentence. Specifically, we added information in the paragraph: “In the case of gynaecological cancers, menopausal status was a determinant factor, especially in breast cancer. In agreement with previous studies, we observed that MS increased breast cancer risk in postmenopausal women and decreased it in premenopausal women. Previous investigations proposed that each component of the metabolic syndrome is connected with systemic alterations. Concerning breast cancer, it has been proposed that components of MS, especially obesity, play different roles in cancer risk according to menopausal status and estrogen receptor status. Obesity is associated with decreased risk of estrogen receptor–positive breast cancer in premenopausal women, but it is closely related with increased risk of estrogen receptor–positive breast cancer in postmenopausal women (ref)”. (page 18, lines 353-360 in the clean copy)

References included: Esposito K, Chiodini P, Colao A, et al: Metabolic syndrome and risk of cancer: a systematic review and meta-analysis. Diabetes Care 2012, 35: 2402-2411. 2012;35:2402-2411. 

Esposito K, Chiodini P, Capuano A, et al. :Metabolic syndrome and postmenopausal breast cancer: systematic review and meta-analysis. Menopause. 2013 Dec;20(12):1301-9. 

Hwang KT, Han KD, Oh S, et al. :Influence of Metabolic Syndrome on Risk of Breast Cancer: A Study Analyzing Nationwide Data from Korean National Health Insurance Service. Cancer Epidemiol Biomarkers Prev 2020, 29: 2038-2047. 

We have also included the following paragraph in the discussion section: “Mechanisms that link metabolic syndrome and cancer risk are not fully understood. Metabolic syndrome may be a surrogate marker for other cancer risk factors, such as decreased physical activity, consumption of high–calorie dense foods, high dietary fat intake, low fiber intake, and oxidative stress (ref).” (page 18, lines 367-370 in the clean copy).

Reference included: Esposito K, Chiodini P, Colao A, et al: Metabolic syndrome and risk of cancer: a systematic review and meta-analysis. Diabetes Care 2012, 35: 2402-2411. 2012;35:2402-2411. 

6. Concerning the important limitation of not following up with the patients, the authors should clearly explain this limitation and how it could affect the results of the paper. 

RESPONSE: There is no follow-up for patients after cancer onset (or index date for controls) in this study. However, data has been treated longitudinally using electronic health records data from healthcare use, which allowed us to determine the timing of exposure in relation to the outcome date. The only limitation in that regard is that we were not able to estimate absolute risks, but only relative risks. We have added a sentence in the paper explaining this limitation “This is a case-control study, when estimating the association between MS and cancer; however, we were not able to estimate cumulative incidences or other types of absolute risks, which would have been useful to put the relative increase in absolute terms”. (page 20, lines 412-415 in the clean copy)

7. Furthermore, the authors should provide information about when the MS component was identified such as previous to tumor diagnose (1 year or 2 years), during tumor treatment…

RESPONSE: Exposure to MS before the event must be at least 2 years prior to cancer diagnosis to avoid reverse causality as it is written in methodological section. We added a phrase in the last paragraph of the “MS definition” section: “Both cases and controls had to be exposed either to MS or to 1 or 2 MS components for at least 2 years before the index date (cancer diagnosis or control identification) to avoid reverse causality” (page 7, line 157 in the clean copy). We did not have information about previous treatment for other health conditions. This is a limitation of our study that we have mentioned in the limitations section. In the paragraph explaining the lack of data, we have added “Also, the lack of data on other possible confounders influencing the relationship between MS and cancer, such as physical activity and parity, or information related to previous treatment for other health conditions might have biased the results” (page 20, line 410 in the clean copy).

Minor comments:

8. A sentence in the abstract is not very clear. “Adjusted conditional logistic regression models were used to estimate OR and 95% CI for the association between individual components of MS and cancer, and MS and cancer.” It seems to be confusing.

RESPONSE: In agreement with the reviewer, we replaced this sentence by the following one: “Adjusted conditional logistic regression models were used to evaluate the association between MS and cancer risk, comparing the effect of global MS versus having one or two individual components of MS”. (Page 2, lines 39-42 in the clean copy)

9. The use of “approximately” when you are talking about the patients available in the study is not accurate for scientific work.

RESPONSE: We agree with the reviewer and have deleted “approximately” accordingly (page 4, line 91 in the clean copy)

10. The authors described in methodology “All individuals ≥ 40-99 years of age with information in the SIDIAP database between 01/01/2006 and 31/12/2017 were included. Final participants included patients with any of the 13 types of incident cancers of interest, together with their paired controls.” Please, the authors should provide the number of patients after the application of each inclusion criteria in the methodology section.

RESPONSE: We agree with the reviewer and have replaced this paragraph with the following: “All individuals ≥ 40 years of age with information in the SIDIAP database between 01/01/2006 and 31/12/2017 were suitable to be included. Patients were excluded from participation when they presented with secondary cancers and metastases.

A total of 190,505 individuals with incident cancer were initially included. Of these, we later excluded 334 men with breast cancer, 6,826 cases because they were diagnosed with more than one cancer on the same day, and 61 because they were >99 years of age on the index date. Finally, a total of 183,284 cases and 733,136 paired controls, four controls for each case, were included (Fig 1).” (pages 4-5, lines 98-105 in the clean copy)

This information was explained at the beginning of the results section, so we have removed it from the results section. We have also removed the word "remaining" from first sentence in the result section so that the sentence is consistent. (page 9, line 214 in the clean copy)

11. Typing errors like “MS component..” should be corrected in the new manuscript.

RESPONSE: The extra period has been removed.

Reviewer #2:

1. In this study, Diana Puente et al studied the potential association between Metabolic syndrome (and number of components met) and the risk of 13 types of cancer. The authors included 183,248 patients from the Information System for Research in Primary Care. The data derived from this study showed that Metabolic Syndrome is associated to 11 cancer types (i.e., endometrial, liver, kidney, pancreas, thyroid, leukaemia, bladder, colorectal, non-Hodgkin lymphoma, lung and post-menopausal breast). These results are interesting for the field and shed light on the relation between Metabolic Syndrome and cancer risk. Therefore, the Reviewer consider that this article is well written, adds to the field valuable information and is suitable to be published in PLOS One. 

RESPONSE: We thank the reviewer for the positive comments.

2. However, the following minor comments should be addressed:

The manuscript is well-written. However, some typos can be found in the text and should be corrected:

o Remove intro in line 130.

o Remove intro in line 131.

o Two period symbols in line 133

RESPONSE: We thank the Reviewer for spotting these typos, we have corrected these specific typos and reviewed the manuscript to make sure that there were no other similar mistakes 

3. In the introduction there is no information about the inconclusive relation between prostate cancer and metabolic syndrome. Authors should include some information about that in the introduction.

RESPONSE: We have included the following sentence in the introduction section with its respective reference: “The results from studies on prostate cancer and MS are inconclusive, while some of them show an increase in risk(Gomez-Gomez E et al.), others show a reduction(Blanc-Lapierre A et al). (page 3, lines 71-73 in the clean copy)

References included: Gomez-Gomez E, Carrasco-Valiente J, Campos-Hernandez JP et al. Clinical association of metabolic syndrome, C-reactive protein and testosterone levels with clinically significant prostate cancer. J Cell Mol Med 2019, 23: 934-942. 

Blanc-Lapierre A, Spence A, Karakiewicz PI et al.: Metabolic syndrome and prostate cancer risk in a population-based case-control study in Montreal, Canada BMC Public Health 2015, 15: 913

4. In addition to the presence of Metabolic Syndrome, the authors also analysed the impact of the number of criteria met on cancer risk. However, no information was shared with regards to the potential association of specific MS criteria and cancer risk. As an example, Gomez-Gomez et al showed that high-blood pressure was associated with clinically significant prostate cancer in their cohort of patients (1). Authors should analyse this or add some information about this in the limitations paragraph.

RESPONSE: This is an interesting issue, and we agree with the reviewer it is important to explore the potential effect of specific components. However, exploring the association of specific MS components and cancer risk was not the main objective of the article (and we did not include it as an objective in the pre-published protocol (Puente D et al.). Additionally, we are providing a comprehensive study looking at the association between single MS components and MS as whole in relation to the risk of numerous types of cancer. The effect of specific components into the risk is likely to vary by cancer type. In order to keep the analyses reasonable for this study, we would prefer to not include this information in the current manuscript. We added some information about this in the limitation paragraph “In addition, it is necessary to explore the potential association of specific MS criteria and risk of specific cancer types in future studies.” (page 20, lines 411-412 in the clean copy).

Published protocol:Puente D, Lopez-Jimenez T, Cos-Claramunt X et al.: Metabolic syndrome and risk of cancer: a study protocol of case-control study using data from the Information System for the Development of Research in Primary Care (SIDIAP) in Catalonia. BMJ Open 2019, 9: e025365

5. The authors performed two sensitivity analyses in which they altered the main definition of Metabolyc Sindrome. The data obtained when using WC instead of BMI are especially interested as WC is a better criterion to ‘measure’ obesity. Although the authors appreciated that the sample is smaller when classifying the patients using WC due to the high number of missing values, this information (number of patients/missing values) is not showed in Supplemental Table 3. Therefore, authors should include this information in the final version of the manuscript.

RESPONSE: We have now included this information at the end of the table “There were 157,872 (86.1%) missing values in the case group and 636,984 (86.9%) missing values in the control group” (page 22, lines 472-743 in the clean copy).

6. The fact that patients with 1 or 2 MS components presented a higher risk of pre-menopausal breast cancer and prostate cancer than participants with MS has not been discussed in the manuscript. Could the authors explain this rare phenomenon? The authors should address this question in the discussion section of the final manuscript.

RESPONSE: Unfortunately, with the data available we could not explore biological mechanisms that could be behind this association. Nevertheless, as indicated in the literature, we found that a study by Gomez-Gomez et al., showed that each of the individual criterion of MS, circulating testosterone levels and inflammatory status may have on the risk and aggressiveness of prostate cancer. Regarding breast cancer, it may be that the presence of a single component could be a risk determinant. Although in our study we cannot determine which component is involved, a study by Xiang Y et al, reported that the hypertriglyceridemic-waist phenotype could be regarded as a strong predictor of breast cancer (ref) and we have included the sentence:” Concerning breast cancer, it has been proposed that components of MS, especially obesity, play different roles in cancer risk according to menopausal status and estrogen receptor status (ref).” (page 18, lines 355-357 in the clean copy) (see response 5 to Reviewer #1)

However, the differences that we find between the presence of 1 or 2 factor and global MS are few and not all of them are statistically significant.

References included: Gomez-Gomez E, Carrasco-Valiente J, Campos-Hernandez JP, et al. Clinical association of metabolic syndrome, C-reactive protein and testosterone levels with clinically significant prostate cancer. J Cell Mol Med 2019, 23: 934-942.

Xiang Y, Zhou W, Duan X, et al. Metabolic Syndrome, and Particularly the Hypertriglyceridemic-Waist Phenotype, Increases Breast Cancer Risk, and Adiponectin Is a Potential Mechanism: A Case-Control Study in Chinese Women. Front Endocrinol (Lausanne) 2019, 10: 905.

---

## [Decision Letter · Decision Letter 1]

15 Feb 2022

Association between Metabolic Syndrome and 13 types of Cancer in Catalonia: a matched case-control study

PONE-D-21-30858R1

Dear Dr. Puente,

We’re pleased to inform you that your manuscript has been judged scientifically suitable for publication and will be formally accepted for publication once it meets all outstanding technical requirements.

Kind regards,

Raul M. Luque, PhD

Academic Editor

PLOS ONE

Reviewers' comments:

Reviewer's Responses to Questions

**Comments to the Author**

1. If the authors have adequately addressed your comments raised in a previous round of review and you feel that this manuscript is now acceptable for publication, you may indicate that here to bypass the “Comments to the Author” section, enter your conflict of interest statement in the “Confidential to Editor” section, and submit your "Accept" recommendation.

Reviewer #1: All comments have been addressed

Reviewer #2: All comments have been addressed

2. Is the manuscript technically sound, and do the data support the conclusions?

Reviewer #1: Yes

Reviewer #2: Yes

3. Has the statistical analysis been performed appropriately and rigorously? 

Reviewer #1: Yes

Reviewer #2: Yes

4. Have the authors made all data underlying the findings in their manuscript fully available?

Reviewer #1: Yes

Reviewer #2: Yes

5. Is the manuscript presented in an intelligible fashion and written in standard English?

Reviewer #1: Yes

Reviewer #2: Yes

6. Review Comments to the Author

Reviewer #1: All the question have been accurately and extensively addressed by the authors improving the scientific quality of this article.

Reviewer #2: The authors addressed all the comments and therefore, the Reviewer considers that this study is suitable to be published in PLOS One.

7. PLOS authors have the option to publish the peer review history of their article (what does this mean?). If published, this will include your full peer review and any attached files.

Reviewer #1: No

Reviewer #2: No

---

## [Editor Report · Acceptance letter]

18 Feb 2022

PONE-D-21-30858R1 

Association between Metabolic Syndrome and 13 types of Cancer in Catalonia: a matched case-control study 

Dear Dr. Puente:

I'm pleased to inform you that your manuscript has been deemed suitable for publication in PLOS ONE. Congratulations! Your manuscript is now with our production department. 

Kind regards, 

on behalf of

Dr Raul M. Luque 

Academic Editor

PLOS ONE